# A Characterization and an Evolutionary and a Pathogenicity Analysis of Reassortment H3N2 Avian Influenza Virus in South China in 2019–2020

**DOI:** 10.3390/v14112574

**Published:** 2022-11-21

**Authors:** Tengfei Liu, Yuhao Huang, Shumin Xie, Lingyu Xu, Junhong Chen, Wenbao Qi, Ming Liao, Weixin Jia

**Affiliations:** 1National Avian Influenza Para-Reference Laboratory (Guangzhou), College of Veterinary Medicine, South China Agricultural University, Guangzhou 510642, China; 2Center for Animal Disease Control and Prevention, Dongguan 523128, China; 3Key Laboratory of Zoonosis, Key Laboratory of Animal Vaccine Development, Ministry of Agriculture and Rural Affairs, Guangzhou 510642, China; 4Key Laboratory of Zoonoses Prevention and Control of Guangdong Province, Guangzhou 510642, China

**Keywords:** H3N2, avian influenza virus, reassortment, characterization, evolutionary, pathogenicity, strong virulence, mice, potential threat

## Abstract

Seasonal H3N2 influenza virus has always been a potential threat to public health. The reassortment of the human and avian H3N2 influenza viruses has resulted in major influenza outbreaks, which have seriously damaged human life and health. To assess the possible threat of the H3N2 avian influenza virus to human health, we performed whole-genome sequencing and genetic evolution analyses on 10 H3N2 field strains isolated from different hosts and regions in 2019–2020 and selected representative strains for pathogenicity tests on mice. According to the results, the internal gene cassettes of nine strains had not only undergone reassortment with the H1, H2, H4, H6, and H7 subtypes, which circulate in poultry and mammals, but also with H10N8, which circulates in wild birds in the natural environment. Three reassorted strains were found to be pathogenic to mice, of these one strain harboring MP from H10N8 showed a stronger virulence in mice. This study indicates that reassorted H3N2 AIVs may cross the host barrier to infect mammals and humans, thereby, necessitating persistent surveillance of H3N2 AIVs.

## 1. Introduction

The H3 subtype influenza virus is an extremely important member of the influenza family. Humans, dogs, pigs, horses and other mammals can be infected with the H3 subtype influenza virus, causing symptoms of high fever, cough and diarrhea, which can even be life-threatening in severe cases [1]. Four influenza pandemics have occurred in recorded human history: the Spanish pandemic in 1918, caused by the H1N1 subtype; the Asian pandemic in 1957, caused by the H2N2 subtype; the Hong Kong pandemic in 1968, caused by the H3N2 subtype; and the H1N1-subtype Influenza outbreak in 2009 in Mexico and the United States. Each genome of the influenza virus consists of eight single-stranded negative-strand RNA fragments: HA, NA, PA, PB1, PB2, M, NP and NS [2]. Both the partial gene segments of the 1957 outbreak of the H2N2 Asian pandemic and the 1968 outbreak of the H3N2 Hong Kong pandemic were derived from H3N2-subtype avian influenza viruses [3,4]. The PB1 gene of the 2009 H1N1-subtype influenza virus is derived from the H3N2 human influenza virus that broke out in Hong Kong in 1968, whereas the PB2 and PA fragments are derived from the avian influenza virus. Following its emergence, this novel reassortment of H1N1 swine influenza has caused human and swine infections. In a recent study, researchers found obvious intrasubtypic reassortment between different H3N2 CIVs or CIV and FIV [5]. Reassortment between the H3N2-subtype human influenza virus and H1N1-subtype swine influenza virus has also been common in swine in recent times [6,7]. A subset of the H3N2 swine influenza viruses from the 1998 outbreak in the United States are products of gene reassortment between the human H3N2 and classical swine H1N1 influenza viruses, whereas another subset of reassortments are derived from the genes of human H3N2, classical swine H1N1, and avian viruses [8]. The H3N2 swine influenza (swH3N2) viruses that are currently circulating in pigs in Guangdong Province carry six internal genes from the 2009-pandemic H1N1 virus (pmd09) with the HA and NA genes belonging to the recent human-like lineage of the H3N2 subtype [9]. Although swine influenza viruses evolve at a slower rate than human influenza viruses [10], strains that have undergone reassortment no longer conform to the laws of natural evolution. The frequent close contact between humans and swine facilitates the circulating transmission of the influenza viruses between them. After adaptation within an individual animal, new variant influenza viruses with high replication fitness and strong transmissibility may be generated, which pose a potential threat to human life and health [11].

Waterfowl generally do not develop disease (or present only mild clinical symptoms) after becoming infected with avian influenza viruses, and they are thus the most suitable reservoir hosts for influenza viruses. As more waterfowl are farmed in regions with dense river distributions in China, where intensively farmed waterfowl and farmhouse backyard poultry are often in contact with wild waterfowl, multiple subtypes of the avian influenza virus can easily spread and coexist among them [12]. The H3N2 subtype of the avian influenza virus is widely distributed among wild and farmed waterfowl and does not often cause morbidity. Domestic ducks, the largest waterfowl in animal husbandry, play an important role in AIV ecology by providing an ideal environment for the reassortment of H3-subtype influenza viruses with other subtypes of influenza viruses. As part of surveillance, H3N2-subtype AIVs can be isolated from some of the incident chicken, duck, and goose flocks that occur on farms, which are likely to be a major cause of the morbidity in farmed animals. In theory, avian influenza viruses only circulate among birds; however, they often mutate, which results in the H5 and H7 subtypes, that had previously only circulated in birds, but are now able to infect humans and even cause multiple deaths [13,14]. The avian-origin H3N2 influenza viruses are able to directly cross the host barrier to infect companion dogs and cause severe clinical symptoms, and they may then infect humans via transmission [15]. A novel reassortment influenza virus was isolated from the nasal swab of a symptomatic cat in Jiangsu Province, China and subsequent sequence analysis indicated the presence of seven genes sharing the highest similarity with the avian-origin canine influenza viruses (CIV H3N2) isolated in China, and an NS gene with sequence identity, indicating it is closely related to the circulating human influenza virus (H3N2) in the region [16]. In a previous study, reassortment was observed between the PB2 segments of the H5N6 and H3N2 subtypes that may have caused pathogenic changes in the H5N6 epidemic branch [17], responsible for the reduced protective efficacy of the extant veterinary vaccine. All six segments of an H9N8 virus strain identified in Korea were found to belong to the H3 subtype, and the reassorted H9N8 virus could replicate in both the respiratory and intestinal tracts of chickens [18]. An H3N2 AIV isolated in domestic ducks in China has the highest sequence homology to the H7-subtype AIVs in all seven segments, except for the HA gene. Although less pathogenic in chickens, it was able to replicate in mouse lungs [19]. According to these studies, the H3N2 subtype is highly susceptible to reassortment with other subtypes of influenza viruses and both the antigenicity and the pathogenicity of the influenza viruses are altered after reassortment, potentially inducing immune escape and the recurrence of an influenza pandemic. Thus, there is a need for surveillance with continuous sampling to determine the reassortment events and possible changes in the pathogenicity to mammals of H3-subtype AIVs.

## 2. Materials and Methods

### 2.1. Sample Collection and Virus Isolation

A total of 672 oropharyngeal and cloacal swabs were collected from chickens, ducks, geese and pigeons in live poultry markets and commercial poultry farms between 2019 and 2020. Each sample was placed in 1 mL of cold phosphate-buffered saline (PBS) containing penicillin (2000 U/mL) and streptomycin (2000 U/mL). After mixing and centrifugation at 10,000× *g*/min for 5 min, 0.2 mL of supernatant was used to inoculate 9-day-old specific-pathogen-free chicken embryos via the allantoic cavity followed by incubation at 37 °C for 48–72 h. We then harvested the allantoic fluid. A total of 10 virus strains were isolated: G152, G155, H159, and H157 are of chicken origin; H151, H34, H140, and H144 are of duck origin; G630 is of goose origin; and G188 is of environmental origin (Appendix A).

### 2.2. Whole-Genome Sequencing of AIV Isolates

RNA was extracted from the harvested virus suspensions using the RNeasy Mini Kit (Qiagen, Hilden, Germany), according to the method recommended by the manufacturer, and the whole gene sequence was amplified using two-step RT-PCR and the universal primers reported by Hoffman [20]. The ex Taq premixed enzyme used for the amplification was purchased from Takara reagent company. PCR products of all eight segments of these viruses were subjected to agarose gel electrophoresis, and the target fragments were recovered using the QiAamp Gel Extraction Kit (Qiagen, Hilden, Germany) and sequenced using an ABI3730 DNA analyzer (Shenggong Bioengineering Co., Ltd., Shanghai, China). Data were merged and assembled using Lasergene sequence analysis software based on the National Center for Biotechnology Information (NCBI) virus database (https://www.ncbi.nlm.nih.gov/, accessed on 9 March 2021).

### 2.3. Reference Strains Information

The sequence information of reference strains was download from GenBank (https://www.ncbi.nlm.nih.gov/genbank/, accessed on 9 March 2021) and GISAID (https://platform.epicov.org/epi3/start, accessed on 16 March 2021), with screening of all HA reference strains from the GISAID database between 2000 and 2020. The distribution of the HA reference strains by host was as follows: 358 strains of avian origin (e.g., chickens, ducks, geese, and wild birds); 52 strains of environmental origin; 74 strains from humans and 123 strains from other mammals (90 strains from canines, 6 strains from horses, and 27 strains from swine). The geographic distribution of the HA reference strains included: 239 strains from Asia, 30 strains from Europe, and 339 strains from North America. As for the NA and internal gene cassettes, we used each of the H3N2 genome sequences from this study as queries to perform local BLASTn search with the default parameters, and collected the top 100 gene sequences in the BLAST output.

### 2.4. Phylogenetic Analyses

MAFFT version 7.058 was used to align each of the eight gene segments and eliminate the sequences with less than 95% of the expected segment length. Duplicate sequences in the gene fragment were removed using PhyloSuite [21]. We performed the phylogenetic analysis three times using the maximum likelihood (ML) method in IQ-TREE under the GTR + F + G4 model with 5000 bootstrap replications. High-quality visualization of the phylogenetic data was performed using the Interactive Tree of Life (iTOL).

### 2.5. Estimating Substitution Rates

Based on the phylogenetic topologies obtained and their bootstrap values, we selected a few representative reference sequences and formed eight smaller data sets. TempEst (version 1.5.1) was used to analyze the R^2^ values of the temporal signals and best-fit model in the selected sequences. For estimating the nucleotide substitution rates of all eight segments, we used the Bayesian Markov chain Monte Carlo (MCMC) method offered in the Bayesian Evolutionary Analysis Sampling Trees (BEAST) (v1.10.4c) [22] and a relaxed molecular clock model with uncorrelated log-normally distributed rates and a coalescent Bayesian Skyline. We set the chain lengths to 500 million iterations and performed sampling every 5000 steps to obtain an effective sample size (ESS) ≥200 and convergence was assessed using Tracer (V1.7.1). Time-scaled summary maximum-clade-credibility (MCC) trees with 10 percent for the post-burn-in posterior were created using TreeAnnotator (V1.10.4), and visualized with FigTree (V1.4.4).

### 2.6. Amino Acid Analysis

BioEdit and MEGA 6.0 were used for analysis of sequence format conversion and key amino acid changes, and the online free software NetNGlyc 1.0 Server (https://services.healthtech.dtu.dk/, accessed on 25 March 2021) was used to predict potential glycosylation sites. MegAlign was used to analyze the sequence homologies, and Simplot was used to analyze the recombination of the HA and NA genes.

### 2.7. Pathogenicity Test of BALB/c-Mice

There have been multiple reports of zoonotic infections caused by the reassortment of human influenza viruses and avian-origin H3N2-subtype viruses. To predict the possible threat of the H3N2 avian influenza virus to human health, we performed the mouse experiments primarily to investigate the pathogenicity of the now circulating H3N2-subtype reassortment strains in mammals. For this, we selected two strains from 2019 and two strains from 2020 corresponding to four H3N2-subtype avian influenza viruses from different sources (H159 isolated from chicken, H34 isolated from duck, G630 isolated from geese, G188 isolated from the environment) and assessed the pathogenicity of these strains from different hosts in mammals. These four strains are characterized by different degrees of reassortment, thus, selecting them for the animal experiments allowed us to better compare the pathogenicity of the strains with different degrees of reassortment in mice. The experimental animals were five-week-old BALB/c-mice purchased from Guangzhou Yancheng Biotechnology Co., Ltd. (Guangzhou, China). For the experiment, 50 mice were randomly divided into a total of 5 groups, with 10 mice in each group, including 1 control group. After administering isoflurane via inhalation as mild anesthesia, each mouse was inoculated with 50 µL of the 10^6^ EID_50_ virus solution in the nasal cavity (with PBS solution administered to the control group). The body weight changes in each group were observed for 14 days. Four days after exposure, we euthanized three random mice from each group, and removed the brain, spleen, lungs, and kidneys from each. The organ virus titers were measured using nine-day-old chicken embryos, calculated using the Reed-Muench method [23] and expressed as the average log_10_ EID 50/g ± SD.

## 3. Results

### 3.1. Phylogenetic Analyses of Viral Envelope-Encoding Genes

According to the analysis of sequence identities of HA gene cassette, the identities of the six strains isolated in 2020 ranges between 98.6% and 99.9%, the identities of the four strains isolated in 2019 ranged between 91.4% and 94.1%, and the identities of the strains isolated in 2019 and 2020 ranged between 91.5% and 95.6% (Appendix A). We did not observe the recombination of the HA and NA genes.

According to the phylogenetic tree of the HA gene, 10 strains were of avian and Asian origin, being markedly distant from the human and mammalian branches and from the H3N2-subtype influenza vaccine strains recommended by the WHO for 2021 and 2022 (Figure 1a). According to the phylogenetic analysis, the strains most similar to G152 and G155 were A/EN/Fujian/20754/2016/H3N3 and A/chicken/Guangxi/14185/2017/H3N2, respectively. G188 exhibited the highest sequence homology with A/duck/Hubei/ZYSYF12/2015/H3N6 (Table 1). The six strains, G630, H34, H140, H144, H151, H157 and H159, were similar to A/EN/Guangxi/32828/2017/H3N2 (Appendix A).

According to the homology analysis of the NA gene, the identities of 6 strains in 2020 were 95.6–100%, the identities of 4 strains in 2019 were 91.3–96.0%, and the identities of 10 strains isolated in 2019 and 2020 were 91.0–95.6% (Appendix A). According to the Phylogenetic analysis of the N2 gene, the branch of the N2 gene of the isolated virus strain was composed of the H1, H3, H4, H5, H6, and H7 subtypes (Figure 1b, Table 1). Except for the H5 and H7 subtypes, these subtypes belong to the low-pathogenic avian influenza subtype, which does not readily cause disease and can circulate in birds for long periods. G188, G630 and H34 exhibited the highest homology with A/chicken/Ganzhou/GZ43/2016/H3N2,A/duck/Guangdong/8/30/DGCP036-C/2017/H6N2 and A/duck/Vietnam/HU8//1918/2017/H3N2, respectively (Table 1). G152 and G155 had the highest homology with A/Environment/Jiangxi/47054/2016/H4N2, whereas H140, H144, H151, H157 and H159 exhibited the highest homologies with A/duck/Guangxi/293D21/2017/H1N2 (Figure 2b, Table 1). In conclusion, G630 is closely related to the H6-subtype AIVs, G152 and G155 are closely related to H4-subtype AIVs and H140, H144, H151, H157 and H159 are closely related to H1-subtype AIVs. The homology relationships of all strains are shown in Appendix A.

### 3.2. Phylogenetic Analyses of Internal Gene

According to internal gene homology analysis, the identities among the 10 isolates were as follows: M (91.1–100%); NP (91.0–100%); NS (92.9–100%); PA (89.7–99.9%); PB1 (88.9–100%); PB2 (87.3–99.1%) (Appendix A). The M-gene sequences of G152, G155 and G188 exhibited the highest homology with A/duck/Mongolia/619/2019/H3N6, A/duck/Shanghai/SH1/2013/H3N2 and A/duck/China/322D22/2018/H3N2, respectively (Appendix A). The NP and NS genes of G152 and G155 exhibited the highest homology with A/duck/China/322D22/2018/H3N2, whereas G188 had the highest similarity to A/chicken/Ganzhou/GZ157/2016/H3N2. For the PA sequences, G152, G155 and G188 exhibited the highest homology with A/duck/Bangladesh/38827/2019/H11N3, A/duck/Japan/AQ-HE103/2015/H1N2 and A/chicken/Ganzhou/GZ43/2016/H3N2, respectively. The PB1 genes of G152, G155 and G188 showed the highest similarity to A/duck/Guangxi/293D21/2017/H1N2, A/duck/Japan/AQ-HE103/2015/H1N2 and A/duck/Hubei/ZYSYF2/2015/H3N6, respectively. For the PB2 sequences, G152 and G155 were most closely related to A/duck/Zhejiang/6D7/2013/H3N2, whereas G188 exhibited the highest homology with A/chicken/Guangxi/165C7/2014/H3N2. For the G630 strains, M and NP genes had the highest homology with A/chicken/Zhejiang/102622/2016/10/26/H10N8 and A/duck/Jiangxi/22215/2013/H7N3, respectively. The PA, PB1, PB2 and NS genes exhibited the highest homology with A/chicken/Yuhuan/YH14/2016/H1N2. In general, the 2019 isolates are closely related to the with H1 subtype.

According to the phylogenetic analysis, for the strains from 2020, the M genes of H140, H144, H151, H157 and H159 showed the highest homologies with A/duck/China/322D22/2018/H3N2, whereas H34 was most similar to A/duck/Jiangshu/YZ916/2016/H3N2. For the NP sequences, H140, H144, H151, H157 and H159 exhibited the highest homology with A/duck/Hunan/7/2015/H3N6, whereas G34 was most similar to A/chicken/Ganzhou/GZ157/2016/H3N2. The NS genes of H34, H140, H144 and H159 had the highest homology with A/chicken/Ganzhou/GZ157/2016/H3N2, whereas H151 and H157 were most closely related to A/common/teal/Shanghai/NH110923/2019/H1N1. For the PA genes, H140, H144, H151 and H159 exhibited the highest homology with A/chicken/Ganzhou/GZ157/2016/H3N2. H34 and H157 exhibited the highest homology with A/duck/China/322D22/2018/H3N2 and A/common/teal/Shanghai/NH110923/2019/H1N1, respectively. The PB1 genes of H34, H140, H144, H157 and H159 exhibited the highest homology with A/duck/Guangxi/293D21/2017/H1N2, whereas H151 is most similar to A/duck/Mongolia/837/2015/H1N1. For the PB2 sequences, H34, H140, H144 and H159 exhibited the highest homology with A/duck/China/322D22/2018/H3N2, whereas H151 and H157 were most closely related to A/chicken/Bangladesh/40619/2019/H9N2. In general, the PB2, PB1 and MP genes formed three groups in the phylogenetic tree, and the PA, NP and NS genes formed two groups. All six internal genes are closely related to the H1-subtype AIVs (Figure 2a–f).

### 3.3. Analysis of H3N2 AIVs Evolutionary Information

To obtain evolutionary information on the reassortment events of H3N2 AIVs, we estimated the evolutionary rates and obtained Bayesian maximum-clade-credibility (MCC) trees for all eight segments. The evolutionary rate was calculated as 3.447 × 10^−3^ substitutions/site/year (95% highest posterior density (HPD) from 4.0646 × 10^−3^ to 5.9315 × 10^−3^) for HA gene and 4.2913 × 10^−3^ substitutions/site/year (95% highest posterior density (HPD) from 3.7177 × 10^−3^ to 4.8743 × 10^−3^) for the NA gene. For the six inner genes, the PB2 gene had the fastest nucleotide substitution rate, which ranged from 4.0646 × 10^−3^ to 5.9315 × 10^−3^. The evolutionary rate estimated for the MP gene ranged from 1.6106 × 10^−3^ to 2.6038 × 10^−3^, which is considered slow (Figure 3 and Figure 4, Appendix A). In the analysis of the MCC trees of the 10 isolated H3N2 viruses, H3N2 AIVs had close relationship with H1 subtype, and also were related to H2, H4, H6 and H7 subtypes (Appendix A). Significantly, we found that the MP gene of G630 derived from the H10N8 subtype, which is prevalent in wild birds and can cause human morbidity (Figure 3).

### 3.4. Key Amino Acids in Eight Proteins

The HA protein-cleavage-site motif of the 10 isolates is PEKQTR↓GLF, which implies that the AIV strain is a lowly pathogenic to poultry. The six potential glycosylation sites of the nine isolated strains are ^22^NDS^24^, ^38^NGT^40^, ^54^NAT^56^, ^181^NVT^183^, ^301^NGS^303^, and ^499^NGT^501^. G152 lacks the ^22^NDS^24^ glycosylation site. Before 2010, the amino acid sequence at positions 24–26 was NST. However, it was SST IN the strain isolated in 2019 and then mutated to SNT in 2020. The 226Q and 228G of the HA gene is consistent with having the characteristic of avian origin. We did not detect neck deletion at the 63–65 position of the NA gene, which can lead to increased replication rate in mammals [24]. The seven potential glycosylation sites of the NA protein of the nine isolates are as follows: NIT, NNT, NWS, NGT, NAT, NGT and NWS; however, the NNT glycosylation site of G152 changed to NST.

The amino acids that were conserved in all the strains were as follows:253D, 292I, 598T, 627E, 676T, 701D and 714S in PB2 [25,26,27,28,29], 436Y, 577K and 622G in PB1 [30,31,32], 26E, 224S, 343A, 356K and 515T in PA [33,34,35,36], 286A,357Q and 437T in NP [37,38], 274H and 294N in NA [39,40], 156D in M1 [26], and 42S in NS [41] (Appendix A). In the PB2 gene, we detected R389K, A588S and L648I changes, which are found to play critical roles in avian and mammalian adaptation in several strains [27,28]; the amino acid 269S and 677T in PB1 [42], 347D and 383D in PA [34,35], and 30D and 215A in M1 [43], which appeared in all strains, increase viral polymerase activity and mammalian virulence appeared in all strains. The mutation 672L in PA appeared in all strains and is related to airborne transmissibility among chickens [44]. We detected 31N in one strain, which increases the resistance to adamantine [45] (Appendix A).

### 3.5. Pathogenicity Test of H3N2 AIVs on BALB/c-Mice

The serum antibody test results of all the challenge groups were positive. The titer of antiserum of G188, G630, H34 and H159 were 2^4^, 2^5^, 2^4^, and 2^4^, respectively. After challenge, some mice in the G630, H34 and H159 groups died (6 mice of G630 group, 3 mice of H34 group and 5 mice of H159 group had died), and their body weights temporarily decreased (Figure 5). The weights of the mice in the H34 and H159 groups began to recover on the fourth day after infection, and the weights of the mice in the G630 group began to recover on the sixth day after infection (the weights of the mice then decreased on the seventh day due to the feeding conditions). All the mice survived in the G188 group, and their body weights increased. According to the results of the organ-virus-content determination after four days of testing, the G630, H34 and H159 virus strains could replicate in mouse lungs. The load of the G630 strain was the highest in the lungs, and the three strains did not replicate in the other organs of the mice (Figure 5). The serum antibody for G188 was positive, but we did not detected the virus in the organs of the mice on the fourth day after testing.

## 4. Discussion

The human influenza virus H3N2 subtype is the cause of the seasonal influenza in the human population. Because of its susceptibility to mutation, the WHO recommends developing new influenza virus vaccines every year to prevent epidemics of mutant virus strains. In 2021 and 2022, the WHO-recommended H3N2-subtype influenza vaccine strains were A/Cambodia/e0826360/2020 (H3N2) and A/Darwin/9/2021 (H3N2), respectively [46,47]. The mutations include single-nucleotide changes and gene-segment reassortment. The phenomenon of two different viruses infecting the same cell and producing hybrid progeny viruses is called reassortment. The reassortment of viruses could cause the existing vaccines to lose their protective efficacy and change the pathogenicity and transmissibility of the original strain, resulting in a serious threat to public health. The reassortment of the H3N2-subtype avian influenza virus with the human influenza virus has repeatedly caused human influenza outbreaks. Avian influenza viruses for which reassortment with other subtypes of influenza viruses frequently manifest can circulate over wide geographic areas and persist for long periods of time. They do not cause the hosts to exhibit clinical symptoms, or they only cause mild clinical symptoms after infection through birds. They do not have specialized prophylactic measures and treatments, and they are able to chronically coexist in the same animal. The host range of the avian-origin H3N2-subtype influenza viruses is wide, but their pathogenicity and isolation rates are not high. They do not compromise the economic benefits of the farming industry and can therefore easily be ignored. In recent years, the scale of livestock culture has continually expanded. The introduction of breeding animals into new regions, and the increased circulation of animals raised in different regions have expanded the possibility of the co-infection of the same host by different regions through different hosts with different subtypes of influenza viruses. The co-existence of avian influenza viruses from different hosts and regions in the same animal increases their chances of reassortment [48]. Recent years, several reassortment H3N2 AIVs are widely detected in the live poultry markets with drug resistance, and also found H3N2 AIVs bind to human-type receptors and transmit in guinea pigs and ferrets [11,42,49]. In this context, the H3N2-subtype avian influenza viruses that are widespread among waterfowl and can be isolated in poultry may act as vectors that transmit the gene segments of influenza viruses, which would have otherwise been circulating in waterfowl across hosts to poultry through the reassortment with strains circulating in poultry. Novel reassortment strains that cause antigenic changes that result in a reduced protective capacity of the vaccine may also improve the transmissibility and pathogenicity of the strain, and the field isolates binding to the human-type means humans could be infected without pre-adaptation. These demonstrate that the reassortment H3N2 avian influenza viruses pose a clear threat to human health and could lead to avian influenza virus pandemics.

The rate of the evolution of the HA gene of the H3N2 subtype of avian influenza viruses has been increasing every year [50]. In our study, we found a nucleotide substitution rate of six internal gene cassettes of H3N2 were maintained at a low level, even though they had undergone extremely complex reassortments. PB2 gene cassette is correlated with enhanced virulence and host adaptation [51], and it is the fastest evolving segment with a 4.9936 (10^−3^ subs/site/year) mean substitution rate. High evolving rate of PB2 means there are more alterations in virulence and host adaptive capacity of H3N2 AIVs. Changes in the position and number of glycosylation sites on the HA protein are involved in the influenza virus evolution [52]. Of the 10 avian-origin H3N2-subtype influenza viruses isolated in our laboratory, the HA genes had homologies between 91.5–95.6%. Moreover, we found that the glycosylation site on the HA protein of the G152 virus strain is missing ^22^NDS^24^, which may lead to an increased affinity of the virus for the receptor [53,54]. The NA gene homology of the 10 isolates were between 91.0–95.6%, and we did not find neck deletions and resistance mutations. We found that five strains are derived from the H1 subtype, three strains from the H6 subtype, and the remaining two strains from the H3 subtype. According to these results, the majority of the HA and NA genes of the H3N2-subtype group evolved from different subtype combinations, which demonstrates a greater likelihood of emerging avian influenza viruses with completely different biological characteristics.

The heterotrimer of the influenza RNA polymerase protein regulates the influenza virus replication and transcription, and PB1 has both polymerase and endonuclease activities [55]. PB2 is strongly associated with the in vitro binding and the in vivo transcription of the influenza virus [56]. Moreover, PB2 is one of the factors determining whether avian influenza viruses can adapt to mammals [25]. NP is closely associated with the formation of vRNP complexes in influenza virus strains. When mutated, it can affect the binding of the NP protein to the polymerase and subsequently affect the vRNA replication and transcription [57]. In this study, we found that some of the PB1, PB2 and NP segments of the H3N2 viruses are closely related to those of the various influenza-A-virus subtypes, and not only the low-pathogenic avian influenza H1 and H4 subtypes, but also the highly pathogenic H7 subtype. We also observed reassortment of the relatively conserved M and NS genes; the M gene of one strain is closely related to the avian H10 subtype, and the NS genes of three strains may have derived from the avian H1 subtype. The reassortment of an H3N2-subtype avian influenza virus with seasonal influenza caused a large outbreak of human influenza and H9N2, which provides all its internal genes cassettes to the H7N9 virus, was responsible for a new H7N9 pandemic [58]. In this study, we observed that H3N2 could undergo reassortment with several subtypes of AIVs circulating in different hosts, including circulating at the wild bird (H10), at poultry and mammalian interfaces (H1, H2, H4, H6, H7). Our study suggesting that avian origin H3N2 subtype may have transmitted circulating strains from wild birds to the poultry and mammalian with reassortment. Significantly, H10 and H7 AIVs had infected humans and leading to multiple human mortality cases.

We are concerned that the H3N2 subtype AIV could act as a bridge to improve the interaction between the H3N2 subtype of human influenza and the H7-subtype AIVs, resulting in the emergence of novel immune-escape human influenza strains or functionally-human infected AIVs. According to the animal experiments, all mice from the test group tested positive for the antibody. The G630 strain in which the internal gene was completely replaced showed higher pathogenicity in the mice than the H34 and H159 strains in which only a few internal genes were replaced. The body weights of the mice in the G630 and H159 challenge groups and those of the mice in the H34 challenge groups started to recover on day four, with the highest viral load in the lungs of the mice in the G630 group four days after infection. G188 contains an internal gene backbone of avian origin from the H3N2 subtype that could infect mice. However, the body weights did not decrease, and we did not detect virus in the lungs. A reasonable hypothesis is that the H3N2 AIVs have different replication phenotypes in mice. Additionally, it is possible that the replacement of internal genes enhances the virulence of the viral strains in mice and that different gene combinations can be more pathogenic to mice. The reassortment of the HA gene of H3N2 avian influenza virus with human influenza has caused a pandemic, and the HA gene of the recommended vaccine strain is far removed from the avian-origin HA gene branch. This warns us that reassortment of H3N2 avian influenza virus could cause a new round of the influenza virus pandemics and threaten public health.

## 5. Conclusions

In our study, we found that the H3N2 subtype had undergone complex reassortments with various influenza-A-virus subtypes and shared the closest relationship with the H1 subtype. Different reassortment isolates could infect mice without preadaptation, which suggests that the H3N2 avian influenza virus may cross the host barrier to infect humans. Concurrently, we found that the vaccine strains recommended by the WHO are far removed from the branch of the H3N2-subtype AIVs are likely to become infectious. If the H3N2 AIV and human influenza virus undergo reassortment again, then the existing vaccine may not provide effective immune protection. Therefore, the continuous surveillance of the prevalence of the H3N2 avian influenza viruses that circulate in poultry is essential.

## Figures and Tables

**Figure 1 viruses-14-02574-f001:**
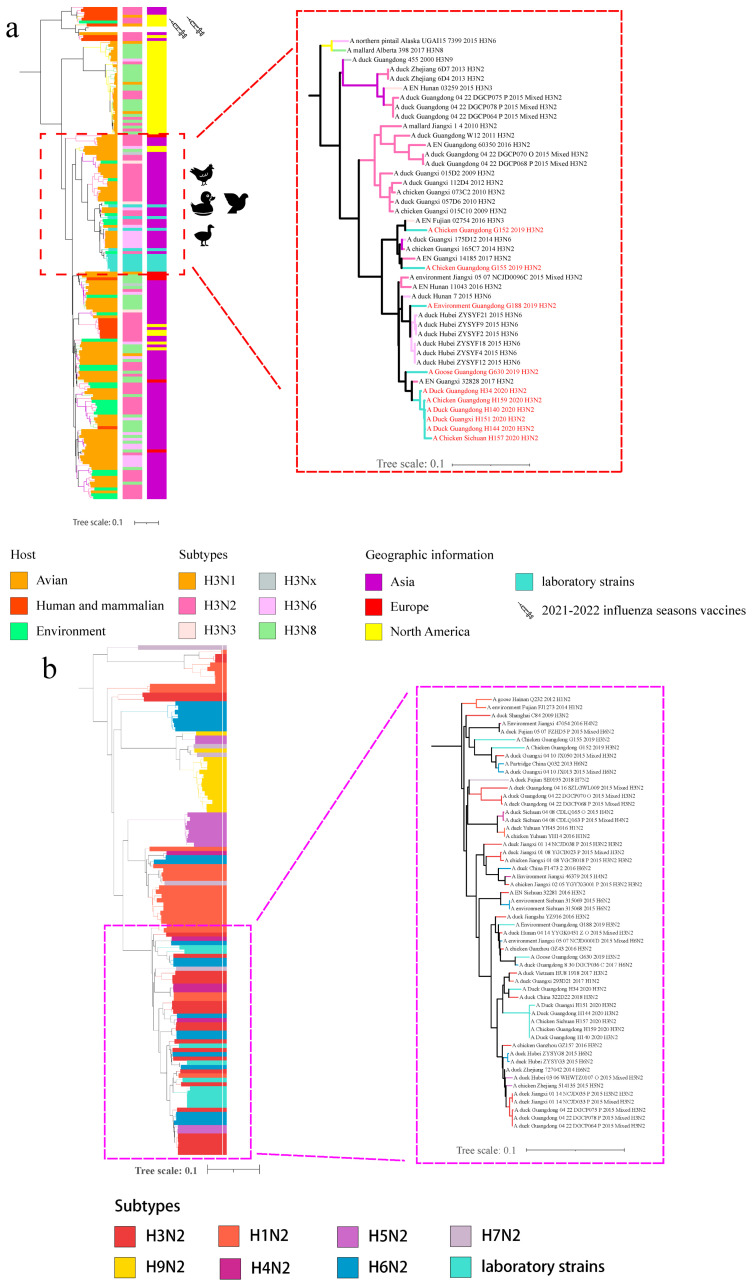
Phylogenetic analysis of HA gene of H3Ny and NA genes of H_X_N2 subtypes. Phylogenetic tree of (**a**) HA gene of H3Ny AIVs; (**b**) N2 gene of HxN2 AIVs. Phylogenetic analysis was performed three times using the maximum likelihood (ML) method in IQ-TREE under the GTR + F + G4 model with 5000 bootstrap replications. Reference sequences were downloaded from the available databases. The phylogenetic tree of the HA was beautified according to the host, subtype and geographic information and that of NA according to the subtype. The turquoise color represents the isolates in this study.

**Figure 2 viruses-14-02574-f002:**
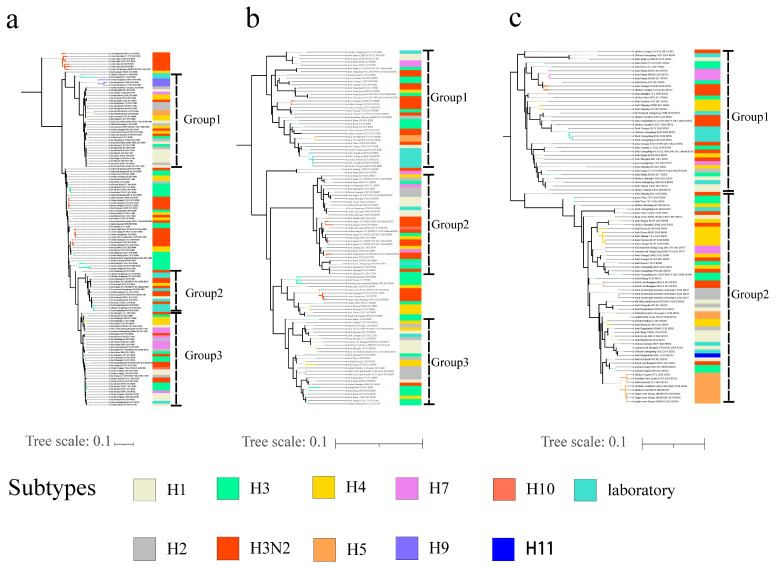
Phylogenetic analysis of internal genes of H3N2-subtype AIVs isolated from 2019 to 2020 using the maximum likelihood method. The phylogenetic trees of PB2 (**a**), PB1 (**b**), PA (**c**), NP (**d**), MP (**e**) and NS (**f**) were obtained as a midpoint-root tree. Phylogenetic analysis was performed three times using the maximum likelihood (ML) method in IQ-TREE under the GTR + F + G4 model with 5000 bootstrap replications. Reference sequences were downloaded from the available databases. The turquoise color represents the isolates in this study. Light yellow2, gray, orange red 1, med spring green, gold, tan 1, violet, slate blue 1, coral 1 and snow 1 represent H1, H2, H3N2, H3, H4, H5, H7, H9, H10, and H11, respectively.

**Figure 3 viruses-14-02574-f003:**
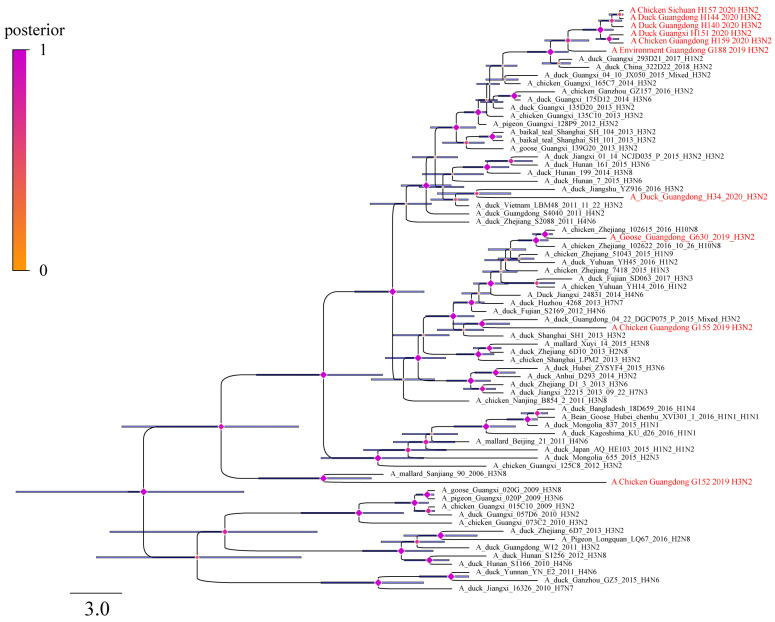
Bayesian Markov chain Monte Carlo method tree of MP gene of isolated H3N2 avian influenza viruses. Blue node bars represent 95% credible intervals of lineage divergence times, and diamonds represent posteriors of every node. The sequence of H3N2 viruses isolated in this study are in red, and viral sequences named in black were downloaded from the databases.

**Figure 4 viruses-14-02574-f004:**
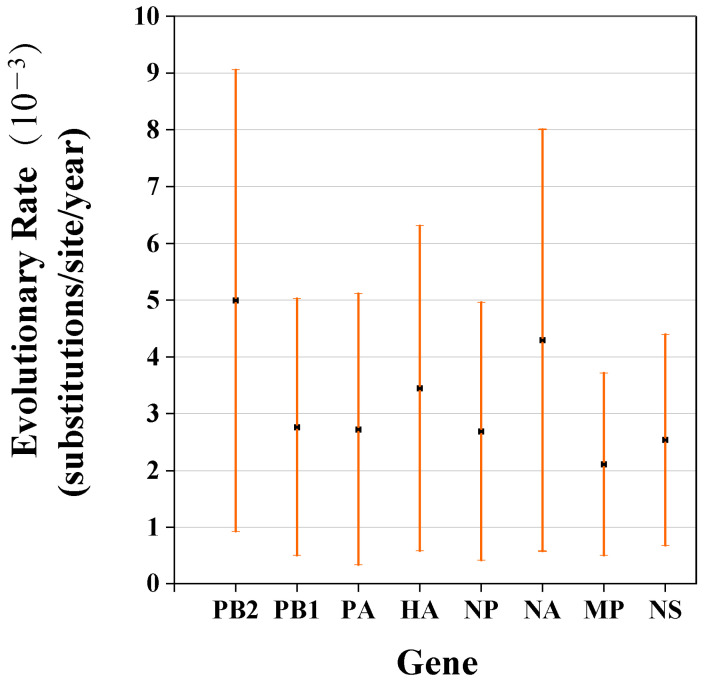
Nucleotide substitution rates for each of the eight segments shown as mean substitution rate for each gene, with the 95% lower and upper HPD values presented as error bars.

**Figure 5 viruses-14-02574-f005:**
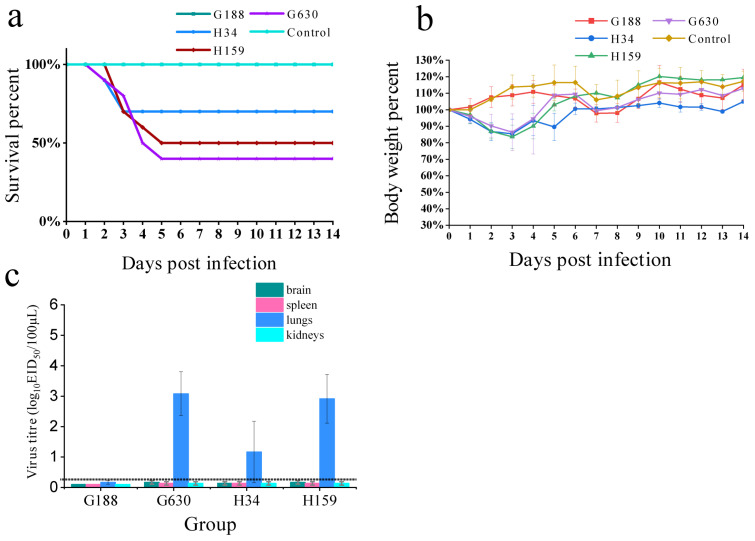
Five-week-old female (SPF) BALB/C-mice were inoculated with 10^6^EID_50_ of the test viruses in a 50 μL volume; (**a**) survival rates of mice (mice that lost more than 20% of their body weights were regarded as dead). (**b**) Changes in body weights of mice; (**c**) horizontal dashed line indicates the lower limit of detection. Each bar represents the virus titer of the four strain replications in the brains, spleens, lungs and kidneys of infected mice.

**Table 1 viruses-14-02574-t001:** Virus strains with the highest homology of each gene segment of the four laboratory isolates.

Strain Name	Gene	Closest Virus	Homology (%)
H34	HA	A/duck/Hunan/7/2015(H3N6)	96.47
	NA	A/duck/China/322D22/2018(H3N2)	98.02
	M	A/duck/Vietnam/LBM48/2011(H3N2)	98.07
	NP	A/chicken/Ganzhou/GZ157/2016(H3N2)	97.8
	NS	A/chicken/Ganzhou/GZ43/2016(H3N2)	98.2
	PA	A/duck/China/322D22/2018(H3N2)	98.03
	PB1	A/duck/China/322D22/2018(H3N2)	97.95
	PB2	A/duck/Guangxi/293D21/2017(H1N2)	98.85
H159	HA	A/duck/Hunan/7/2015(H3N6)	96.36
	NA	A/duck/Zhejiang/727042/2014(H6N2)	96.45
	M	A/duck/China/322D22/2018(H3N2)	99.42
	NP	A/duck/Guangdong/S4040/2011(H4N2)	100
	NS	A/chicken/Ganzhou/GZ43/2016(H3N2)	97.53
	PA	A/chicken/Ganzhou/GZ157/2016(H3N2)	97.81
	PB1	A/duck/Guangxi/293D21/2017(H1N2)	97.74
	PB2	A/duck/Guangxi/293D21/2017(H1N2)	97.78
G188	HA	A/duck/Hubei/ZYSYF18/2015(H3N6)	97.51
	NA	A/chicken/Ganzhou/GZ43/2016(H3N2)	98.08
	M	A/duck/China/322D22/2018(H3N2)	99.32
	NP	A/chicken/Ganzhou/GZ157/2016(H3N2)	97.93
	NS	A/chicken/Ganzhou/GZ43/2016(H3N2)	98.43
	PA	A/chicken/Ganzhou/GZ43/2016(H3N2)	97.54
	PB1	A/duck/Hubei/ZYSYF2/2015(H3N6)	98.3
	PB2	A/chicken/Guangxi/165C7/2014(H3N2)	97.31
G630	HA	A/duck/Hubei/ZYSYF18/2015(H3N6)	95.53
	NA	A/duck/Guangdong/8.30_DGCP036-C/2017(H6N2)	98.09
	M	A/chicken/Zhejiang/102622/2016(H10N8)	99.29
	NP	A/duck/Jiangxi/22215/2013(H7N3)	99.4
	NS	A/chicken/Zhejiang/51048/2015(H1N9)	98.47
	PA	A/chicken/Yuhuan/YH14/2016(H1N2)	97.90
	PB1	A/chicken/Zhejiang/51048/2015(H1N9)	97.5
	PB2	A/duck/Yuhuan/YH45/2016(H1N2)	97.86

## Data Availability

This study did not report any data.

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
