# Peer review of "A Characterization and an Evolutionary and a Pathogenicity Analysis of Reassortment H3N2 Avian Influenza Virus in South China in 2019–2020"

_viruses, 2022, doi:10.3390/v14112574_

Round 1

Reviewer 1 Report

Comments:

In this manuscript, the authors characterized the reassortment H3N2 avian influenza viruses in South China in the period of 2019-2020. In addition to the phylogenetic analysis of the viral genes and evolutionary assessment of the virus, they also tested the pathogenicity of selected strains. Their results suggest that H3N2 avian influenza viruses may serve as a bride between human H3N2 influenza viruses and H7 avian influenza viruses. The data were well presented and interpreted.     

Specific comments:

1.    Can the authors discuss the characteristics, evolution, and pathogenicity of the reassortment H3N2 avian influenza viruses in South China in the context of the reassortment influenza viruses that have been reported? Are there any unique aspects for the viruses that they have studied?  

2.    Nucleotide substitution rates were different among different viral genes. For example, PB2 had a fast rate. Any discussion about this?

Reviewer 2 Report

The authors have presented the manuscript   "Characterization, Evolutionary and Pathogenicity Analysis of Reassortment H3N2 Avian Influenza Virus in South China in 2019 - 2020" the authors have described the findings nicely in a scientific way however there are certain concerns  which are required to be addressed:

1. Line 107 hemocoagulant allantoic fluid need to be changed to allantoic fluid.  Lines 130-131 are not clear kindly rewrite. line 181 calculating should be replaced with calculated. In results section lines 193-197 in phylogenetic analysis how authors are writing about homology it should be either viruses grouped or genetically related to, kindly check this throughout results section. What is the similarity percent it has to be specified. Nucleotide substitution rates in supplementary files are not matching as presented in results for HA gene. Substitution  rates of PB2 gene are higher which need to be verified and discussed. Table 3 and Fig3 may be corrected in text as per the data presented. Line 323 some mice died it has to be specified.. Line 332 serum antibody titre if available should be  presented.
